# Salsolinol-Containing *Senna silvestris* Exerts Antiviral Activity Against Hepatitis B Virus

**DOI:** 10.3390/plants14152372

**Published:** 2025-08-01

**Authors:** Alberto Quintero, Maria Maillo, Nelson Gomes, Angel Fernández, Hector R. Rangel, Fabian Michelangeli, Flor H. Pujol

**Affiliations:** 1Laboratorio de Virología Molecular, Centro de Microbiología y Biología Celular, Instituto Venezolano de Investigaciones Científicas, Caracas 1020A, Venezuela; albertojosequinteroaraque@gmail.com (A.Q.); hrangel2006@gmail.com (H.R.R.); 2Laboratorio de Genética y Reproducción Animal, Centro de Biotecnología Agrícola, Instituto Venezolano de Investigaciones Científicas, Caracas 1020A, Venezuela; 3Departamento de Química Medicinal, Instituto Venezolano de Investigaciones Científicas, Caracas 1020A, Venezuela; arrakisfirst@yahoo.com; 4Unidad de Química Aplicada, Centro de Química, Instituto Venezolano de Investigaciones Científicas, Caracas 1020A, Venezuela; 5Laboratorio de Fisiología Gastrointestinal, Centro de Biofísica y Bioquímica, Instituto Venezolano de Investigaciones Científicas, Caracas 1020A, Venezuela; angelfern56@yahoo.com (A.F.); fmichelangeli@gmail.com (F.M.)

**Keywords:** hepatitis, antiretroviral, plant extracts, antiviral, salsolinol

## Abstract

Several natural products have been shown to display antiviral activity against the hepatitis B virus (HBV), among a number of other viruses. In a previous study, the hydro-alcoholic extracts (n = 66) of 31 species from the Venezuelan Amazonian rain forest were tested on the hepatoma cell line HepG2.2.15, which constitutively produces HBV. One of the species that exerted inhibitory activity on HBV replication was *Senna silvestris*. The aim of this study was the bioassay-guided purification of the ethanol fraction of leaves of *S*. *silvestris*, which displayed the most significant inhibitory activity against HBV. After solvent extraction and two rounds of reverse-phase HPLC purification, NMR analysis identified salsolinol as the compound that may exert the desired antiviral activity. The purified compound exerted inhibition of both HBV DNA and core HBV DNA. Pure salsolinol obtained from a commercial source also displayed anti-HBV DNA inhibition, with an approximate MIC value of 12 µM. Although salsolinol is widely used in Chinese traditional medicine to treat congestive heart failure, it has also been associated with Parkinson’s disease. More studies are warranted to analyze the effect of changes in its chemical conformation, searching for potent antiviral, perhaps dual agents against HBV and HIV, with reduced toxicity.

## 1. Introduction

Around half of the pharmaceutical drugs are derived from natural products [1]. Several natural products have been shown to display antiviral activity against the hepatitis B virus (HBV), among a number of other viruses. In spite of the availability of a highly effective vaccine, hepatitis B is still a significant health concern in the world: around 300 million people are chronic carriers of this virus [2]. HBV is a hepadnavirus, with a genome composed of partially double-stranded DNA generated from an intermediate RNA through reverse transcription [3]. Two main types of drugs are currently available against HBV chronic infection: pegylated interferon (PegIFN-α) and some antiretroviral drugs, particularly some reverse transcriptase inhibitors. However, a cure after treatment for HBV infection is rarely achieved [3].

The main goal of an effective HBV treatment is to achieve the functional cure, defined as the elimination of the HBV surface antigen (HBsAg), in the absence of HBV DNA [4]. For this, three main categories of targets can be described [4,5]:-Inhibition of HBV replication. In addition to PegIFN-α and nucleos(t)ide analogs, other antivirals, such as entry inhibitors, capsid assembly modulators (CAMs), short interfering RNAs (siRNAs), and antisense oligonucleotides (ASOs), are evaluated as potential candidates.-Inhibition of HBsAg production or secretion. PegIFN-α, siRNA, ASO, and second-generation CAMs can inhibit the production of this antigen, while nucleic acid polymers can inhibit secretion.-Finally, the functional cure could be achieved through the restoration or stimulation of the innate immune response (TLR7 or TLR8 agonists), and of the adaptive response (therapeutic vaccines), passive immunization (antibodies to HBsAg), and the removal of immune blockade (by attacking the immune checkpoint inhibitors). PegIFN-α, as an immune modulator, also plays a role in this critical aspect.

The effective cure for HBV will probably consist of a combination of treatments against these three targets, with a tailored-made protocol for each patient [4,5].

In the era of highly active antiretroviral therapy, liver-related diseases have emerged as a significant cause of morbidity and mortality among patients with HIV. As for HIV-1, there is an urgent need for new and more effective anti-HBV compounds. Natural products offer the opportunity to identify some of these compounds. Several plants used as medicines by indigenous communities have shown antiviral activities. Many of them have complementary and overlapping mechanisms of action [6].

In a previous study, the hydro-alcoholic extracts (n = 66) of 31 species from the Venezuelan Amazonian rain forest were tested on the hepatoma cell line HepG2.2.15, which constitutively produces HBV [7]. One of the species that exerted inhibitory activity on HBV replication was *Senna silvestris*. The aim of this study was the bioassay-guided purification of the ethanol fraction of leaves of *S. silvestris*, which displayed the most significant inhibitory activity against HBV.

## 2. Results

In a previous work, we showed that the leaf extract of *S. silvestris* exhibited antiviral activity against HBV in a concentration-dependent manner in HepG2.2.15 cells, which constitutively express HBV [7]. This extract was not cytotoxic at 100 µg/m in both HepG2 and HepG2.2.15 cells [7]. A semi-quantitative PCR detection system was standardized to evaluate the inhibition of HBV replication in HepG2.2.15 cells by each plant extract [7]. In order to detect inhibitory effects at different steps of the HBV replication cycle, HBV DNA from core particles, cccDNA from the cellular nucleus, and reverse transcriptase activity were determined. The *S. silvestris* leaf extract exerted an inhibition of core HBV DNA generation, showing a concentration- and time-dependent effect, clearly observed at 9 days of incubation (Figure 1). However, this extract did not induce a reduction in cccDNA nor affect the viral reverse transcriptase activity (less than 10% reduction).

A bioassay-guided fractionation was performed on *S. silvestris* leaf extract in order to determine the active compound(s) responsible for antiviral activity. For this purpose, a preliminary chemical characterization was made in order to investigate the principal components of *S. silvestris* leaves. The chemical analysis of the extract showed the presence of polyphenols and flavonoids (Table 1). By solvent fractionation, the ethanol fraction (2.5 g) exhibited the highest antiviral activity among all of them (Figure 2A). In addition, this fraction inhibited the accumulation of core HBV DNA in a time and concentration-dependent manner (Figure 2B). RP-18 silica column fractionation was then performed, using solvents of decreasing polarity, from water to chloroform. Only the first fraction eluted with water (969 mg), displayed antiviral activity (34.3 ± 3.6 percent inhibition on HBV DNA production).

Reverse-phase HPLC was then performed on Fraction 1 using a water–methanol gradient program. From the 15 main peaks obtained, F6 (34.3 mg) displayed the most important antiviral activity (Figure 3 and Table 2). NMR and GC-MS analysis of F6 revealed that this fraction was not pure.

A further purification step was performed by programming a more progressive polarity gradient to obtain a better compound separation of F6, which yielded four different peaks, with G3 (8.6 mg) being the one exhibiting antiviral action (Figure 4 and Figure 5 and Table 3). The estimated IC50 values for HBV DNA and core HBV DNA were 49 and 29 µM, respectively, with no apparent cytotoxicity as detected by microscopic observation of the cells.

NMR and GC-MS were performed on peak G3 in order to elucidate the chemical structure of the antiviral compound. The ^1^H and ^13^C NMR data revealed the presence of a known compound: 1-metil-6,7-dihidroxy-tetrahydroisoquinoline (1-methyl-6,7-dihydroxy-1,2,3,4-tetrahydroisoquinoline), better known as salsolinol (SAL) (Table 4 and Table 5 and Appendix A). The spectroscopic data were similar to those reported previously for this compound [15,16]. Mass spectrometry evaluation indicated that there was a 164.06 g/mol compound. The molecular weight (MW) of SAL is 179 g/mol. This type of compound loses a methyl group, remaining in a more stable conformation, when subjected to mass spectrometry, and is compatible with this result (Figure 6). In addition to SAL, the spectroscopic data are also compatible with the presence of Nor-salsolinol (Figure 6).

Commercial SAL (Sigma^®^, Ronkonkoma, NY, USA) was then evaluated for antiviral activity. Anti-HBV activity was confirmed for SAL in a dose-dependent manner, with an approximate MIC value of 0.01 µg/mL, around 12 µM (Figure 7). From the data, it cannot be excluded that Nor-salsolinol may also exhibit antiviral activity.

## 3. Discussion

*S. silvestris* belongs to the second largest genus of the subfamily *Caesalpinioideae*, family *Leguminosae*, with about 300 species and a pantropical distribution. Of these, over 20 species have been reported to have traditional medical uses for the treatment of a wide range of diseases and infections. *Senna* spp. have been investigated due to the structural diversity of bioactive molecules coupled with a broad spectrum of biological and pharmacological activities exhibited by aerial and underground parts of *Senna* species (root, bark, stem, leaf, seed, and fruit) [17].

Phytochemical investigations of *Senna* spp. have revealed over 120 structurally diverse compounds, which include alkaloids, terpenoids, glycosides, tannins, saponins, steroids, flavonoids, anthraquinones, polyphenols, and anthrones [18,19]. Furthermore, pharmacological studies have confirmed that crude extracts, fractions, or isolated metabolites of the genus *Senna* possess antimalarial, antidiabetic, antimicrobial, antioxidant, anti-inflammatory, analgesic, antitumor, antinociceptive, and anticancer properties [17,19].

The extract of *S. silvestris* leaves was active against core DNA formation, and W. coccinea leaves had inhibitory action against cccDNA generation [7]. CAMs are promising molecules in the search for HBV treatment [4,5,20]. Because of the ability to inhibit this intermediate in the HBV life cycle, extracts of *S. silvestris* leaves were chosen for bioassay-guided purification.

The purification strategy consisted of a bioactivity-guided (inhibition of DNA HBV and core DNA formation) fractionation scheme. First, gross fractionation using solvents of different polarity, followed by separation on an open column using RP-18 silica gel, and, finally, isolation of compounds by HPLC. Analyses of the chemical structure revealed that G3 and G4 fractions, derived from the second HPLC purification step, were biogenic amines commonly found in plants and animals. The anti-HBV activity was present only in G3. This purified fraction appeared to contain one compound, which was determined by NMR and GC-MS analysis to be 1-methyl-6,7-dihydroxy-1,2,3,4-tetrahydroisoquinoline, commonly known as SAL. The analysis did not discard the presence of another compound, Nor-salsolinol. As mentioned before, we cannot discard that Nor-salsolinol also may exert antiviral activity against HBV. However, commercial pure SAL has been confirmed to be active against HBV.

SAL is a catecholamine derivative from a non-enzymatic condensation of acetaldehyde with dopamine [21]. The N-SAL, a methyl derivative of SAL, is highly toxic in neuroblastome dopaminergic cells, interfering with cellular metabolism [21,22]. Alternatively, SAL can be oxidized in the presence of hydrogen peroxide to form melanine [23]. SAL can be found in different beverages and foods like ripe bananas and other rotten fruits, beef, chocolate, cheese, and milk [24,25]. To the best of our knowledge, this is the first report of SAL being found in plants of the genus *Senna*.

Iwasa et al. [26] determined that tetrahydroisoquinoline and 6,7-dihydroxyisoquinolium salts derived from SAL exhibit antiviral activity against HIV-1, with an IC_50_ of 0.117 μg/mL. This IC_50_ is similar to the one found against HBV. However, they did not provide further evidence on the mechanism of action of the compound. HIV-1 and HBV share some critical steps in their replication cycle. As mentioned previously, SAL did not seem to exert any effect on the HBV reverse transcriptase. We might then speculate that SAL might also exert its effect on HIV-1 through inhibition of capsid assembly.

SAL was first detected in humans in the urine [27] and the cerebral spinal fluid (CSF) [28] of patients with Parkinson’s disease. Multiple hypotheses have been proposed for a neurotoxic role of this molecule. One recent study suggests the role of SAL in RNA m6A methylation to induce neuronal death [29]. Epidemiological studies show that alcohol intake is an important risk factor for breast cancer. It has been suggested that SAL, as a metabolic product of alcohol, may be responsible for this association by inducing DNA damage and mammary cell proliferation [30]. On the other hand, SAL is widely used in Chinese traditional medicine to treat congestive heart failure and has been proven in vitro to improve angiotensin II-induced myocardial fibrosis [31]. Moreover, recent studies suggest a neuroprotective effect of enantiomers of SAL and one methyl derivatives [27].

SAL is an example of the complexity of evaluating new antiviral compounds because of the difficulty of evaluating their long-term effects for the treatment of chronic infections. However, the discovery of anti-HBV, and probably also anti-HIV activity, suggests a promising horizon to search for SAL derivatives, retaining its antiviral activity and abrogating its neurotoxic and other oxidative effects. Studies are warranted to analyze the effect of modifications in its chemical conformation, searching for potent antiviral, perhaps dual agents against HBV and HIV, with reduced toxicity.

## 4. Materials and Methods

### 4.1. Isolation of Antiviral Compounds

Leaves of *S. silvestris* (collected by B. Milano and B. Andrews, # collection 507, in Yutajé, Municipio Manapiare, Amazonas State, Venezuela, on 20 January 1997) were collected and extracted, as previously described by [7]. Briefly, the plants were collected and extracted fresh in a field laboratory. After grinding in a blender, the plant material was macerated in three volumes of 95% ethanol for at least three days. The material was filtered, and the alcoholic extract was concentrated and dried by rotavaporation and lyophilization. The dried powder was stored at −80 °C until use. Stock extract solutions (100 mg/mL) were prepared in 50% ethanol.

The first step of the separation of *S. silvestris* leaf extract was performed using 20 g of lyophilized extract dissolved in water. Extraction was performed with a series of solvents: hexane, chloroform, ethyl acetate, acetone, ethanol (Fluka Riedel-de Haen), and distilled water. Each fraction was evaluated for antiviral activity. The resulting fraction with inhibitory capacity (2.5 g) was loaded onto an RP-18 silica gel column and eluted with decreasing polarity gradient from water to CHCl_3_ (chloroform). Fifteen fractions eluted from the RP-18 column were assayed for their antiviral activity. Later, the anti-HBV fraction (962.9 mg) was further purified by means of reverse phase HPLC in a Waters ^®^ 4000 controller, Millipore, and a Water^®^ 486 detector using a 5 mm diameter RP-18 column (50 × 250 mm). The mobile phase was a 20 min gradient from 0 to 40% acetonitrile in water at a flow rate of 4 mL/min and detection at 240 nm. Each peak was evaluated for antiviral activity. Finally, the inhibitory compound (10 mg) was studied for its chemical structure.

### 4.2. Structural Analyses

Electron impact mass spectra were performed on a HITACHI high-performance spectrophotometer. ^1^H NMR, ^13^C NMR, COSY, HMBC, and HMQC spectra were obtained on a Bruker AVANCE 500 MHz spectrometer in DMSO and D_2_O (^1^H NMR only).

### 4.3. Cells and Cytotoxicity Assays

All cellular lines were grown in RPMI supplemented with fetal bovine serum (FBS). After reaching confluence, the hepatoma cell line HepG2.2.15 was cultivated in the presence of 1% dimethyl sulfoxide (DMSO) without FBS to favor viral production. HepG2 and HepG2 2.2.15 cells were incubated with different concentrations of lyophilized plant extract (10, 100, and 1000 µg/mL). Viability was determined using the MTS test after 6 days (for HepG2 and HepG2-derived cells) of incubation with the plant extract, as described previously [7]. Extracts were considered cytotoxic when reducing the viability of the cell lines to less than 80%.

### 4.4. Testing of Inhibitory Activity on HBV Replication

HepG2.2.15 cells were seeded at 25,000 cells/well in T24 plates, incubated with or without plant extracts. Supernatants were collected every 3 days until day 9 and stored at −70 °C. Inhibition of viral DNA production was detected by semi-quantitative PCR, as previously reported [7].

For the determination of covalently closed circular DNA (cccDNA), HepG2.2.15 cells were seeded as described previously in T-25 culture flasks. After the extraction of viral DNA from cells, cccDNA was detected as described by Lu et al. [32], which consists of the partial digestion with mung bean nuclease of the single-strand region of the non-covalently closed HBV DNA, while cccDNA remains intact for amplification with primers directed to this region of the genome.

For the determination of DNA inside HBV core particles, the cells were seeded in T-24 plates until confluence. Extracts with antiviral activity were inoculated at 100 µg/mL, and the medium was replaced every three days until day six, when they were collected. Cells were lysed with hypotonic solution in the presence of detergent (50 mM Tris-HCl, 1 mM EDTA, 1% Nonidet P-40, pH 7.4) for 5 min, then placed on ice for 15 min and centrifuged at 14,000 rpm for 1 min. Supernatant (40 µL) was digested with RQ1 DNase according to manufacturer instructions (Promega Corporation, Fitchburg, WI, USA) at 37 °C for 1 h. The resulting viral DNA was extracted and amplified as described above.

HBV reverse transcriptase activity was measured using the EnzChek Reverse Transcriptase Assay (Thermo Fisher Scientific, Waltham, MA, USA). The virus in the supernatant of HepG2.2.15 grown without fetal bovine serum and 1% DMSO was used for the assay, in the presence and absence of the extract.

## Figures and Tables

**Figure 1 plants-14-02372-f001:**
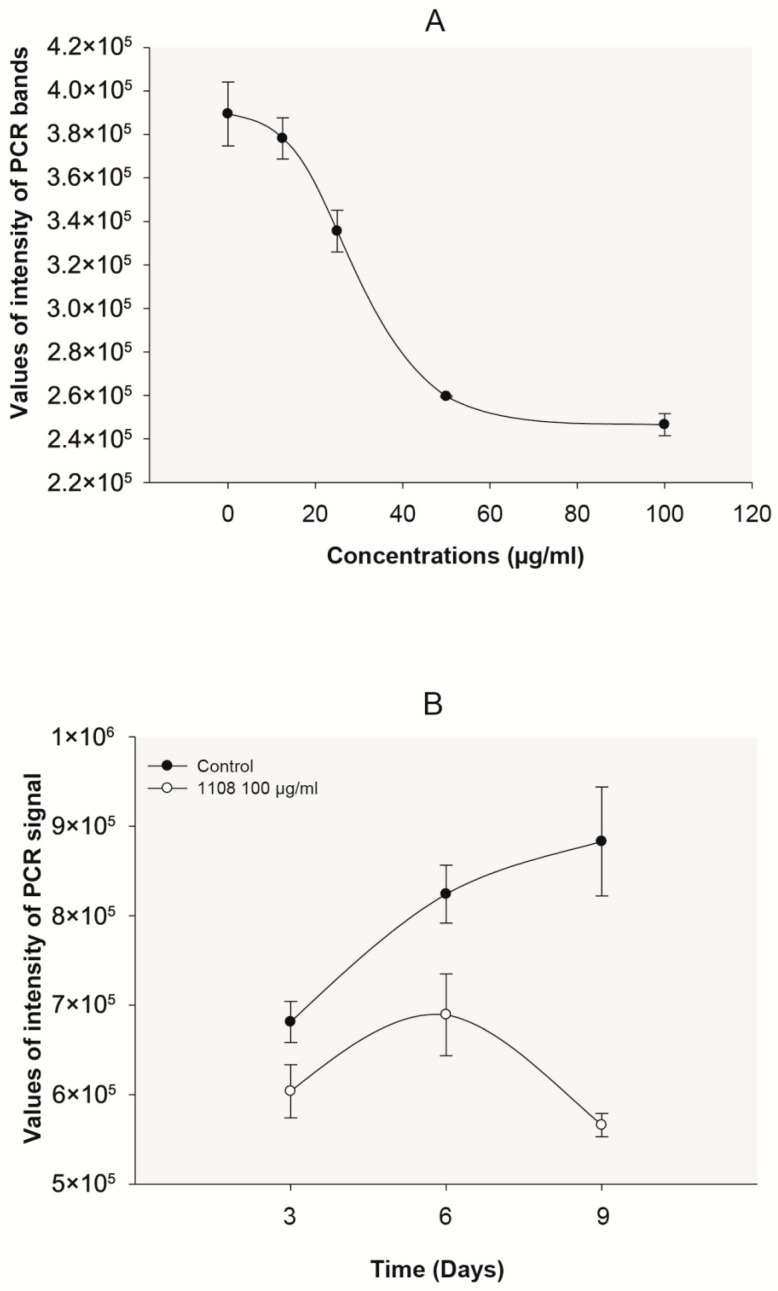
Inhibition of core HBV DNA found in HepG2.2.15 cells with extracts of *S. silvestris* leaves at different concentrations (**A**) and at different days of incubation (**B**). Fresh culture medium with or without extract was changed every three days until day 9. The PCR signals correspond to a cumulative effect (n = 3 replicas). Concentration-dependent curve (**A**) was performed at day 6.

**Figure 2 plants-14-02372-f002:**
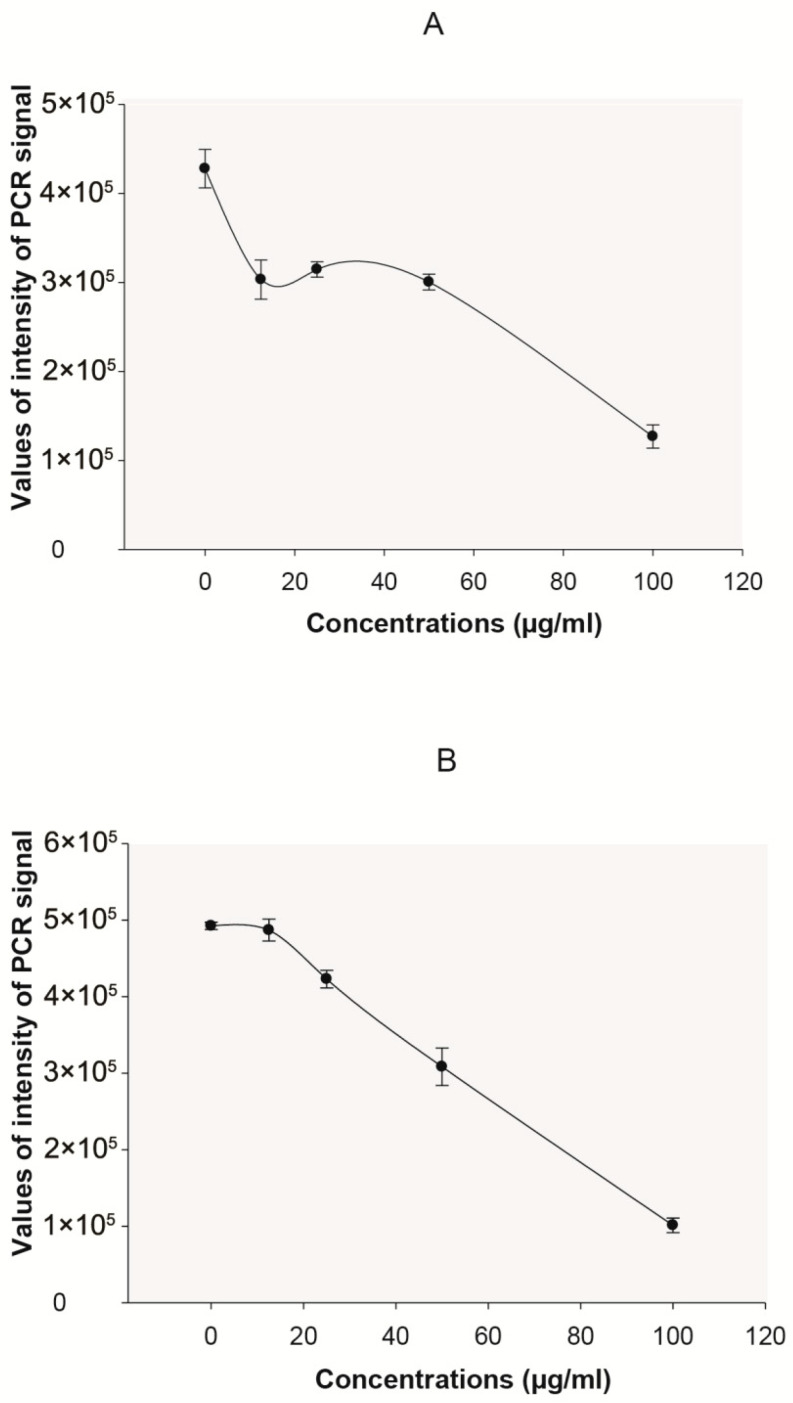
Inhibition assays of HBV DNA in HepG2.2.15 cells with ethanolic fraction from leaves of *S. silvestris* at different concentrations. (**A**) Inhibition assay at day 6. The PCR signals correspond to a cumulative effect (n = 3 replicas). Concentration-dependent curve was performed at day 6. (**B**) Inhibition assay of core DNA HBV at day 6. The PCR signals correspond to a cumulative effect (n = 3 replicas).

**Figure 3 plants-14-02372-f003:**
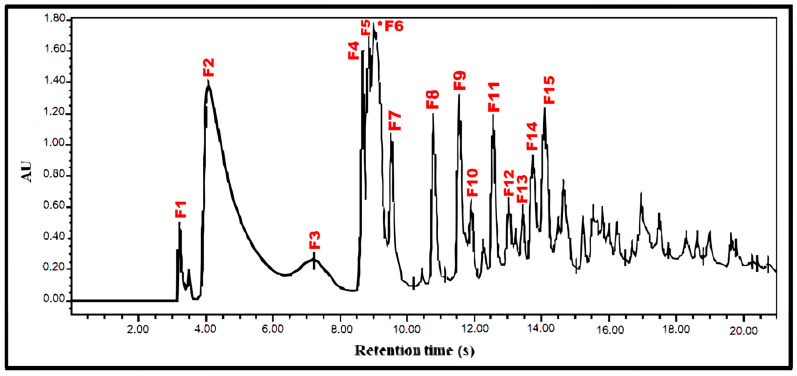
Chromatogram corresponding to the separation of Fraction 1 (F1) by running it in an HPLC-UV system. The sample was diluted ½ and injected (150 μL) into an RP-C18 column for 20 min. Gradient program was CNMetOH from 0 to 40% for 20 min. The peaks were detected by absorbance at 240 nm. The asterisk indicates Peak 6 (F6), responsible for antiviral activity by * Peak 6 (F6), responsible for antiviral activity.

**Figure 4 plants-14-02372-f004:**
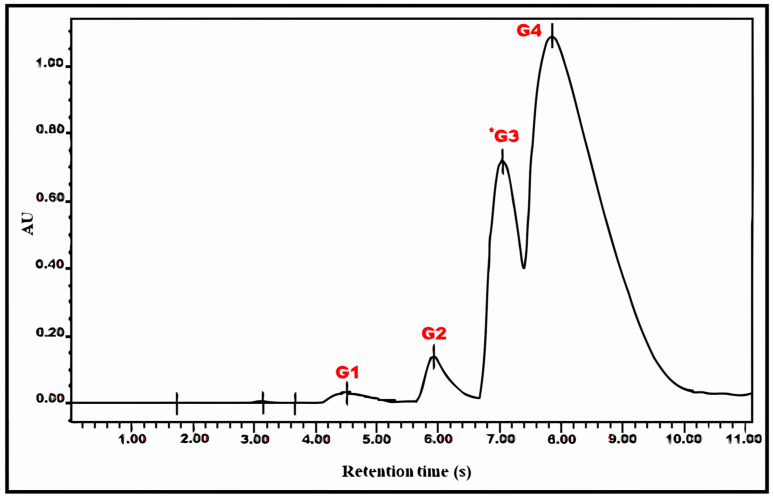
Chromatogram corresponding to the separation of peak 6 (F6) by running it in an HPLC-UV system. The sample was injected (150 μL) into an RP-C18 column for 20 min. The gradient program was CNMetOH from 0 to 20% for 20 min. The peaks were detected by absorbance at 240 nm. The asterisk indicates Peak 3 (G3), responsible for antiviral activity by * Peak 3 (G3), responsible for antiviral activity.

**Figure 5 plants-14-02372-f005:**
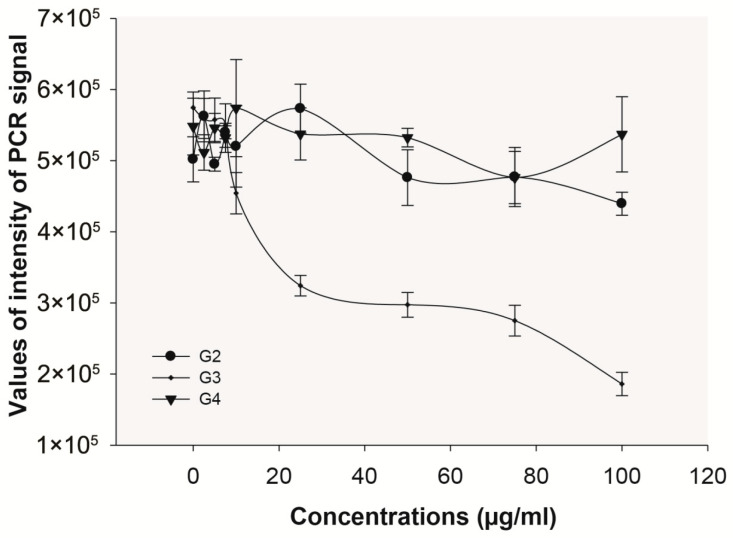
Inhibition assays of HBV DNA in HepG2.2.15 cells with different fractions (G2, G3, and G4) proceeding from F6 separation employing an HPLC-UV system, at different concentrations. The PCR signals correspond to a cumulative effect (n = 3 replicas). Concentration-dependent curve was performed at day 6. The PCR signals correspond to an accumulative effect (n = 3 replicas).

**Figure 6 plants-14-02372-f006:**
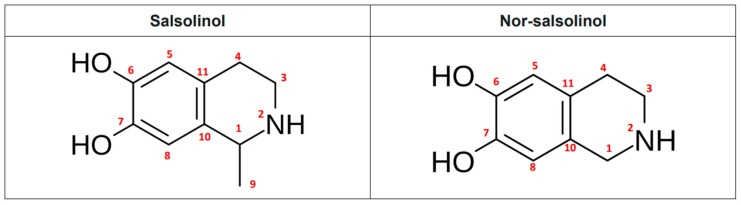
Chemical structure of 1-metil-6,7-dihidroxytetrahydroisoquinoline (SAL), as the molecule responsible for the antiviral activity against HBV. The chemical structure of Nor-salsolinol is also shown.

**Figure 7 plants-14-02372-f007:**
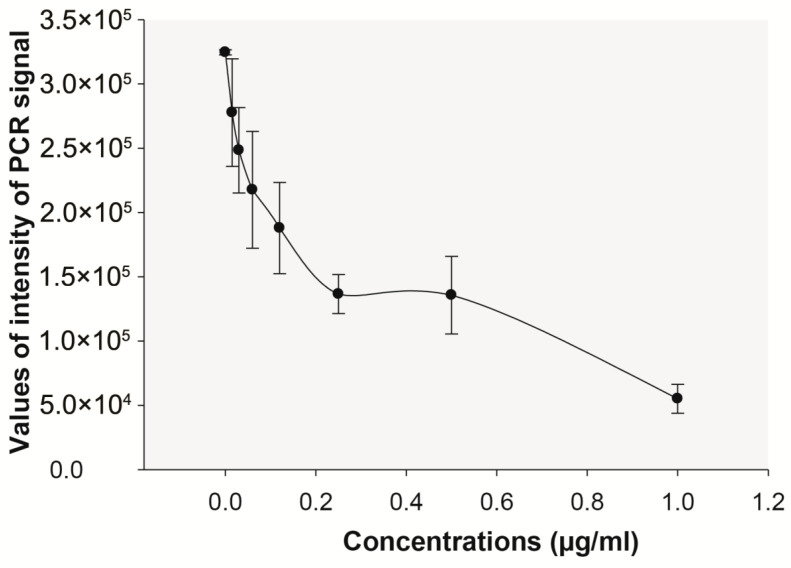
Inhibition assays of HBV DNA in HepG2.2.15 cells by SAL at different concentrations at day 6 of incubation. Fresh culture medium with or without SAL was changed every three days. The PCR signals correspond to a cumulative effect (n = 3 replicas). Concentration-dependent curve was performed at day 6.

**Table 1 plants-14-02372-t001:** Chemical analysis of the *S. silvestris* extract.

Chemical Analysis	Compound Detected	Result
FeCl_3_	Polyphenols	Positive
Shinoda	Flavonoids	Negative
Jelly	Tannins	Negative
Dragendorff	Alkaloids	Positive
Afrosymetric index	Saponins	Negative
Libermann-Bouchard	Triterpenes	Negative

The compounds were detected according to the following procedures: polyphenols [8], flavonoids [9], tannins [10,11], alkaloids [12], saponins [13], and triterpenes [14].

**Table 2 plants-14-02372-t002:** Inhibition assay for FI separation through HPLC-UV running, showing the %RS (percent relative PCR signal) for the 15 major peaks.

Fractions	Retention Time	Mass	% RS
(100 μg/mL)	(min)	(mg)	
F1	3.486	2.3	113 ± 4.22
F2	4.074	32.6	103 ± 3.0
F3	7.21	3.4	140 ± 5.7
F4	8.666	3.4	100 ± 3.4
F5	8.878	2	65.8 ± 2.9
**F6 ^1^**	**9.002**	**34.3**	**64.1 ± 3.1**
F7	9.539	14.5	82.9 ± 4.1
F8	10.799	4.4	114 ± 2.8
F9	11.583	5.2	127 ± 4.0
F10	11.92	3	111 ± 3.9
F11	12.569	1	120 ± 2.5
F12	13.039	12	108 ± 2.0
F13	13.452	6.2	142 ± 3.8
F14	13.753	1	121 ± 4.5
F15	14.102	7.3	117 ± 2.6

^1^ All the peaks were assayed at 100 μg/mL except peak 6 (F6, in bold), which was evaluated at 10 μg/mL. At this concentration, the cytotoxic activity of F6 was minimal (21%).

**Table 3 plants-14-02372-t003:** Inhibition assay for F6 separation through HPLC-UV running, showing the %RS (percent relative PCR signal) for the four major peaks.

Fractions	Retention Time	Mass	%RS
100 µg/mL	(min)	(mg)	
G1	4.509	0	0
G2	5.93	3.2	96.9 ± 3.2
**G3 ^1^**	**7.053**	**8.6**	**28.42 ± 4.6**
G4	7.852	20.4	96.31 ± 3.5

^1^ All the peaks were assayed at 100 μg/mL. Peak 3, G3 (in bold) exerted the highest inhibitory activity.

**Table 4 plants-14-02372-t004:** Spectroscopic data of the G3 peak, purified compound obtained after a second HPLC of the F6 peak: ^1^H NMR.

N	Type of Proton	δ (ppm)	Salsolinol	Nor-Salsolinol
**H1**	CH	4.46 (q, J ≈ 6.8 Hz)	✔	✘ (CH_2_)
**H9**	CH_3_	1.54 (d, J ≈ 6.8 Hz)	✔	✘
**H3**	CH_2_	3.49–3.28 (m)	✔	✔
**H4**	CH_2_	2.94–2.88 (m)	✔	✔
**H5**	Aromatic CH	6.69 (s)	✔	✔
**H8**	Aromatic CH	6.72 (s)	✔	✔

δ (chemical shift, ppm), J (coupling constant), q (quartet), d (doublet), m (multiplet), and s (singlet). ✘: absent.

**Table 5 plants-14-02372-t005:** Spectroscopic data of the G3 peak, purified compound obtained after a second HPLC purification of the F6 peak: ^13^C NMR.

N	Type of Carbon	δ (ppm)	Salsolinol	Nor-Salsolinol
**C1**	CH (chiral)	50.4	✔	✘ (CH_2_ ≈ 45 ppm)
**C9**	CH_3_	19.6	✔	✘
**C3**	CH_2_	40.1	✔	✔
**C4**	CH_2_	24.8	✔	✔
**C5**	Aromatic CH	115.5	✔	✔
**C6**	Aromatic C-OH	145.4	✔	✔
**C7**	Aromatic C-OH	144.74	✔	✔
**C8**	Aromatic CH	113.2	✔	✔
**C10**	Aromatic CH	124.7	✔	✔
**C11**	Aromatic CH	122.3	✔	✔

δ (chemical shift, ppm), J (coupling constant), q (quartet), d (doublet), m (multiplet), and s (singlet). ✘: absent.

## Data Availability

The original contributions presented in this study are included in the article/Appendix A. Further inquiries can be directed to the corresponding author.

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
