# Peer review of "Salsolinol-Containing Senna silvestris Exerts Antiviral Activity Against Hepatitis B Virus"

_plants, 2025, doi:10.3390/plants14152372_

Round 1
Reviewer 1 Report
Comments and Suggestions for Authors
This manuscript is of average quality in the results, additionally should be corrected.
All figures have mistakes, please check them (in a yellow shadow in the manuscript attached).
Why did you include two different spectroscopic data for the same compound?
What wavelength did you use in the detector for separation in HPLC?
Figure 1 has been reported in a previous publication, could you move it to the introduction?
You mention in the manuscript "data no shown", could you show them in the supplementary material?
Please improve the redaction of your manuscript

Author Response
Reviewer 1
This manuscript is of average quality in the results, additionally should be corrected.
All figures have mistakes, please check them (in a yellow shadow in the manuscript attached).
We apologize for the mistake in the symbol in the Figures, that seems to occur in some versions of the manuscript but not in the one we sent. This is now corrected. The strange symbol means µg/ml.
Why did you include two different spectroscopic data for the same compound?
Two spectroscopic data were included: the one obtained in this study and the one previously reported by Francisco et al. (ref 15). This explanation was added in the legend of Figure 6 and in lines 162-163 of page 7. Small differences in values are expected because of differences in calibration between different equipment.
What wavelength did you use in the detector for separation in HPLC?
The wavelength used for detection in HPLC was 240, as mentioned in Materials and Methods. The wavelength used is now also included in Figures 3 and 4.
Figure 1 has been reported in a previous publication, could you move it to the introduction?
We apologize for the duplication. The Figure 1 is now eliminated in the revised version and substituted by the reference of the paper.
You mention in the manuscript "data no shown", could you show them in the supplementary material?
The mention of ¨data not shown¨ was substantially reduced in the manuscript, in lines 81, 130-131.
Please improve the redaction of your manuscript
The manuscript has been thoroughly edited for English accuracy.
Reviewer 2 Report
Comments and Suggestions for Authors
1.The authors need to provide the MIC value of salsolinol against viruses, as well as toxicity data for both the crude plant extract and this compound.
2.The manuscript contains several non-standard expressions, such as "礸/mL", "(d, J = 6.8, H9)"1H NMR (500.13 MHz) (D2O). Please carefully verify and revise the whole manuscript.
3.The authors should provide the carbon numbering for compound salsolinol in Figure 7. The author should rewrite the content of Line 152-159, in page 7.
4.I suggestthe author to present data in figures using bar charts instead of current formats.
Comments on the Quality of English Language
1.The authors need to provide the MIC value of salsolinol against viruses, as well as toxicity data for both the crude plant extract and this compound.
2.The manuscript contains several non-standard expressions, such as "礸/mL", "(d, J = 6.8, H9)"1H NMR (500.13 MHz) (D2O). Please carefully verify and revise the whole manuscript.
3.The authors should provide the carbon numbering for compound salsolinol in Figure 7. The author should rewrite the content of Line 152-159, in page 7.
4.I suggestthe author to present data in figures using bar charts instead of current formats.
Author Response
1.The authors need to provide the MIC value of salsolinol against viruses, as well as toxicity data for both the crude plant extract and this compound.
An approximation of the MIC value for Salsolinol is now included in page 8, lines 181-182, from Figure 7, in the revised version. The cytotoxicity assay results in HepG2 and HepG2.2.15 cells of the crude extract was included in our previous publication and then is mentioned in lines 73-74 and cited. Cytotoxicity evaluation of the purified compound is now included in text in lines 130-132, 142 (footnote of Table 2). For the purified compound, the cytotoxicity was only evaluated by visual inspection of cells at the inverted microscope, in sake of preserving the purified compound for antiviral analysis.
2.The manuscript contains several non-standard expressions, such as "礸/mL", "(d, J = 6.8, H9)"1H NMR (500.13 MHz) (D2O). Please carefully verify and revise the whole manuscript.
We apologize for the mistake in the symbol in the Figures, that seems to occur in some versions of the manuscript but not in the one we sent. This is now corrected. The strange symbol means µg/ml. The other expressions were corrected and defined as well (see legend of Figure 6).
3.The authors should provide the carbon numbering for compound salsolinol in Figure 7. The author should rewrite the content of Line 152-159, in page 7.
The carbon numbering is now provided in the new figure 6.
4.I suggest the author to present data in figures using bar charts instead of current formats.
We apologize but we prefer the standard curve format used in the original version.
Reviewer 3 Report
Comments and Suggestions for Authors
The main question addressed by this research is whether the plant Senna silvestris, which has previously shown inhibitory activity against hepatitis B virus (HBV) replication, contains specific compounds responsible for this antiviral effect. The study aims to identify and characterize these compounds through bioassay-guided purification and determine their potential as antiviral agents. The originality of this work lies in its focused investigation of Senna silvestris as a source of antiviral compounds against HBV. While previous studies have identified antiviral potential in various plant extracts, this study provides a detailed purification and identification of the specific compound, salsolinol, responsible for the observed antiviral activity. This work addresses a significant gap in the field by exploring natural products from understudied plant species in the Venezuelan Amazonian rainforest as potential sources of new antiviral agents, particularly against HBV, which remains a global health challenge despite the availability of vaccines.
This study adds valuable insights into the specific antiviral compounds present in Senna silvestris. The identification of salsolinol as an active compound against HBV is novel and provides a basis for further research into its mechanism of action and potential therapeutic applications.
The methodology used in this study, including bioassay-guided fractionation and structural analysis, is robust and appropriate for the research objectives.
The manuscript is reasonably well-written; however, the following issues require attention before proceeding further.
- About the research method,several improvements could be considered. First,the study could benefit from additional controls,such as including known antiviral compounds as positive controls in the inhibition assays to validate the assay system. Second, he authors could consider conducting time-course studies to better understand the kinetics of salsolinol's antiviral activity. Third, the study could include more detailed cytotoxicity assays to fully assess the potential side effects of salsolinol and its derivatives. Finally, the authors might consider exploring the mechanism of action of salsolinol in more depth in the discussion, such as through studies on its effects on viral entry, replication, or assembly.
- The references provided are generally appropriate and relevant to the study. They cover the background on HBV, the use of natural products as antiviral agents, and the pharmacological properties of salsolinol. However, the authors could consider including more recent studies on HBV treatment strategies and the latest advancements in antiviral research to provide a more comprehensive context for their work.
- Units of measurement must use standardized symbols. What does "礸" mean in the tables/figures? Please correct it.
- Latin names in references should be italicized, e.g., Senna in reference #15. Check all references thoroughly.
- The manuscript contains numerous technical errors, raising concerns about its adherence to rigorous scientific standards. Examples include formatting issues with "CHCl3" and "IC50."
Author Response
The main question addressed by this research is whether the plant Senna silvestris, which has previously shown inhibitory activity against hepatitis B virus (HBV) replication, contains specific compounds responsible for this antiviral effect. The study aims to identify and characterize these compounds through bioassay-guided purification and determine their potential as antiviral agents. The originality of this work lies in its focused investigation of Senna silvestris as a source of antiviral compounds against HBV. While previous studies have identified antiviral potential in various plant extracts, this study provides a detailed purification and identification of the specific compound, salsolinol, responsible for the observed antiviral activity. This work addresses a significant gap in the field by exploring natural products from understudied plant species in the Venezuelan Amazonian rainforest as potential sources of new antiviral agents, particularly against HBV, which remains a global health challenge despite the availability of vaccines.
This study adds valuable insights into the specific antiviral compounds present in Senna silvestris. The identification of salsolinol as an active compound against HBV is novel and provides a basis for further research into its mechanism of action and potential therapeutic applications.
The methodology used in this study, including bioassay-guided fractionation and structural analysis, is robust and appropriate for the research objectives.
The manuscript is reasonably well-written; however, the following issues require attention before proceeding further.
Thank you.
- About the research method,several improvements could be considered. First, the study could benefit from additional controls, such as including known antiviral compounds as positive controls in the inhibition assays to validate the assay system. Second, he authors could consider conducting time-course studies to better understand the kinetics of salsolinol's antiviral activity. Third, the study could include more detailed cytotoxicity assays to fully assess the potential side effects of salsolinol and its derivatives. Finally, the authors might consider exploring the mechanism of action of salsolinol in more depth in the discussion, such as through studies on its effects on viral entry, replication, or assembly.
Positive controls (lamivudine) were included in the first steps of evaluation (of the 66 plant extracts), to validate the assays, as well as the cytotoxicity assay results in HepG2 and HepG2.2.15 cells. This last information is included in our previous publication, and is cited in text (lines 73-74). Cytotoxicity evaluation during the purification steps is included in text, lines 130-131, line 142.
- The references provided are generally appropriate and relevant to the study. They cover the background on HBV, the use of natural products as antiviral agents, and the pharmacological properties of salsolinol. However, the authors could consider including more recent studies on HBV treatment strategies and the latest advancements in antiviral research to provide a more comprehensive context for their work.
We have included a description of the path to a functional cure for HBV infection (lines 41-57) with two additional references.
- Units of measurement must use standardized symbols. What does "礸" mean in the tables/figures? Please correct it.
We apologize for the mistake in the symbol in the Figures, that seems to occur in some versions of the manuscript but not in the one we sent. We hope this is now corrected. The symbols is µg/l. The other expressions were corrected as well.
- Latin names in references should be italicized, e.g., Senna in reference #15. Check all references thoroughly.
The Latin names were revised and edited.
- The manuscript contains numerous technical errors, raising concerns about its adherence to rigorous scientific standards. Examples include formatting issues with "CHCl3" and "IC50."
We apologize for the mistakes. We edited and corrected throughout the manuscript.
Reviewer 4 Report
Comments and Suggestions for Authors
See the revision file

No comments
Author Response
We hope to have improved the manuscript in the revised version.
Reviewer 5 Report
Comments and Suggestions for Authors
Senna silvestris contains salsolinol, which exerts antiviral activity against hepatitis B virus. Comments: This is perhaps an interesting work, but it needs some important revisions.
- Include in the abstract data on the activity of the fraction and the compound salsosinol.
- There is no information on the species in pharmacological, phytochemical, etc. studies.
- In a previous work, we showed that the leaf extract of S. silvestris exhibited antiviral activity against HBV in a concentration-dependent manner in HepG2.2.15 cells, which constitutively express HBV.
- Describe how the structure was determined using two-dimensional NMR (COSY, NOESY, HSQC, and HMBC). Explain how you determined that nitrogen was present. 15N NMR must be obtained. 5. Check all spelling and especially the 1H and 13C NMR chemistry.
6.- Place a table with the 1H and 13C NMR data and add the one- and two-dimensional NMR data as supplementary data. Or the FIDS.
- The methodology data is incomplete and does not describe the fraction that was subjected to reverse-phase column analysis. A significant range of compounds appears in the mass spectrometry analysis, and only one was isolated. I believe there is very little phytochemical information, and it is a known compound.
Overall, I believe your document lacks a research format with results, statistical analyses, and negative and positive controls. It should have a sequence and order. The document is difficult to read. The chemical elucidation of Salsosinol is not understandable.
Comments on the Quality of English LanguageNone
Author Response
Senna silvestris contains salsolinol, which exerts antiviral activity against hepatitis B virus. Comments: This is perhaps an interesting work, but it needs some important revisions. Include in the abstract data on the activity of the fraction and the compound salsosinol. There is no information on the species in pharmacological, phytochemical, etc. studies.
We included in the Abstract information on activity of the fraction and of pure commercial salsolinol. There is no information available in the literature on phytochemical studies of Senna silvestris. Phitochemical studies of the genus were already included in Discussion (second paragraph) in the previous version of the manuscript. In addition, the chemical analysis of the leaves extract is included in Table 1.
In a previous work, we showed that the leaf extract of S. silvestris exhibited antiviral activity against HBV in a concentration-dependent manner in HepG2.2.15 cells, which constitutively express HBV. Describe how the structure was determined using two-dimensional NMR (COSY, NOESY, HSQC, and HMBC). Explain how you determined that nitrogen was present. 15N NMR must be obtained. 5. Check all spelling and especially the 1H and 13C NMR chemistry. 6.- Place a table with the 1H and 13C NMR data and add the one- and two-dimensional NMR data as supplementary data. Or the FIDS.
We have no access to two-dimensional NMR. However, our NMR data is similar to the one previously published by other authors for salsolinol (see ref 15 and 16: a new reference is now included). Thanks to the comments of the reviewer, we performed a more detailed analysis of the NMR data and found that in addition to salsolinol, nor-salsolinol may also be present. We include this information in page 7, lines 165-166. However, since commercial pure salsolinol was evaluated and found to exert anti-HBV activity, we are confident that this compound in the fraction has antiviral activity against HBV. We cannot exclude that nor-salsolinol can also have anti-HBV activity: this information was included in page 8, lines 181-182.
The methodology data is incomplete and does not describe the fraction that was subjected to reverse-phase column analysis. A significant range of compounds appears in the mass spectrometry analysis, and only one was isolated. I believe there is very little phytochemical information, and it is a known compound.
We apologize for the lack of clarity in the presentation of the purifications steps. Two rounds of HPLC were performed. From the first HPLC purification step, the fraction F6 was selected as the one containing antiviral activity, and then subjected to a second round HPLC, with a more progressive polarity gradient, which lead to peak G3, which contained a purified (or more purified) compound with the antiviral activity. This information was described in more detail in the revised version (page 5, lines 128-130).
Overall, I believe your document lacks a research format with results, statistical analyses, and negative and positive controls. It should have a sequence and order. The document is difficult to read. The chemical elucidation of Salsosinol is not understandable.
We hope that the new version (after two rounds of revision) is better organized and more understandable.
Reviewer 6 Report
Comments and Suggestions for Authors
In their manuscript, Quintero et al. reports an experimental study investigating the presence of active compounds of Senna silvestris leaves using bioassay-guide purification, where one of them salsolinol (SAL) exerts antiviral activity against hepatitis B virus. Based on their results, the authors highlight the possibility that modified SAL and SAL derivates may be used as potent antiviral agents against HBV.
The manuscript presents original and high quality scientific data. The results reported in the manuscript are original and represent an important scientific contribution to the field. The manuscript is well written, the results are presented clearly, methods are state-of the art.
I suggest minor improvement:
Methodology for the measurement of viral reverse transcriptase activity should be clarified.
Author Response
Reviewer 6
Round 1
In their manuscript, Quintero et al. reports an experimental study investigating the presence of active compounds of Senna silvestris leaves using bioassay-guide purification, where one of them salsolinol (SAL) exerts antiviral activity against hepatitis B virus. Based on their results, the authors highlight the possibility that modified SAL and SAL derivates may be used as potent antiviral agents against HBV.
The manuscript presents original and high quality scientific data. The results reported in the manuscript are original and represent an important scientific contribution to the field. The manuscript is well written, the results are presented clearly, methods are state-of the art.
I suggest minor improvement:
Methodology for the measurement of viral reverse transcriptase activity should be clarified.
We thank the reviewer for the comments and brought some information on the measurement of reverse transcriptase activity in Methodology, at the end of page 11.
Round 2
Reviewer 1 Report
Comments and Suggestions for Authors
You made changes in the manuscript; however, I find some mistakes still.
In Figure 1B. 1108 100 µg/mL.
In page 3-Line 101, specie in cursives
On page 5- lines 123 and 124 (data not shown) appear yet.
In Figures 3 and 4, Gradient program was CNMetOH from 0 to 40 % for 20 min.
In Tables 4 and 5 the “assignment” column is confusing, I recommend that you delete it.
Author Response
You made changes in the manuscript; however, I find some mistakes still.
In Figure 1B. 1108 100 µg/mL.
We apologize for the mistake, now corrected.
In page 3-Line 101, specie in cursives
Thank you, now corrected (line 98).
On page 5- lines 123 and 124 (data not shown) appear yet.
The sentence was eliminated, since the inhibition of HBV core is shown later in Figure 5, for the purified F6 fraction.
In Figures 3 and 4, Gradient program was CNMetOH from 0 to 40 % for 20 min.
We apologize, but the second gradient was 0-20% as stated.
In Tables 4 and 5 the “assignment” column is confusing, I recommend that you delete it.
Tables 4 and 5 were modified, and the assignment deleted, as suggested.
Reviewer 2 Report
Comments and Suggestions for Authors
There are still some errors in the manuscript. The author should checked the manuscript carefully.

No
Author Response
There are still some errors in the manuscript. The author should checked the manuscript carefully.
The manuscript was again thoroughly revised, and some errors were edited. Thank you.
Reviewer 3 Report
Comments and Suggestions for Authors
The authors have demonstrated considerable diligence and sincerity in their revisions to the manuscript,which has led to a marked improvement in its overall quality.I am pleased to recommend that the manuscript be accepted for publication.
Author Response
Thank you.
Reviewer 4 Report
Comments and Suggestions for Authors
File in PDF

File in PDF
Author Response
Communication
Plants-3615639
Senna silvestris contains salsolinol, which exerts antiviral activity against hepatitis B virus
Results
The scientific name of Senna silvestris must be in italics in all of the manuscript
The scientific names are all in Italic in the revised version.
- The NMR H1 and C13 data revealed the presence of a known compound. H1 and
13C must be in superindex
We apologize for the mistake, now corrected in page 7, line 152.
- The authors compared the previously reported NMR data of salsolinol with their
experimental results. However, the signal -CH2-N is missing in 1H and 13C
experimental data. The absence of these signals indicates that the described
compound does not correspond with an alkaloid.
Francisco, M.C.; Nasser, A.L.; Lopes, L.M. Tetrahydroisoquinoline alkaloids and 2-
deoxyribonolactones from Aristolochia arcuata. Phytochemistry, 2003, 62, 1265-
- doi: 10.1016/s0031-9422(02)00655-6.
I recommend:
- Verifying the identity of the main compound before a resubmission process
We wish to thank the reviewer for this comment. We have thoroughly revised the NMR data and could confirm the presence of salsolinol. A new table is included. In addition, we cannot discard the presence of nor-salsolinol. However, the antiviral activity found with pure commercial salsolinol (Figure 7) confirms that salsolinol exerts anti-HBV activity. This data is commented in page 7, lines 165-166. We cannot exclude that nor-salsolinol can also have anti-HBV activity: this information was included in page 8, lines 181-182.
Round 2
Considering that this work aims to show the biological activity of an alkaloid from Senna
silvestris, it is necessary to confirm the chemical structure of this naturally occurring
compound. For this reason, I would like to revise the 13C and 1H NMR experiments. They
could be added to the supplementary material.
We added two tables (4 and 5) on this data. We hope that we addressed correctly in this version the concerns.
There is no concordance between the 1H and 13C signals in Figure 6 and the numbering of the chemical structure. The authors described chemical shifts for C-9, C-10, and C-11.
However, they did not appear in the figure.
We corrected Figure 6 and included nor-salsolinol.
In addition, the signal for C-3 is missing in both the first and second versions of the
manuscript.
The data is now included in Table 5.
Reviewer 5 Report
Comments and Suggestions for Authors
The document was improved, but it doesn't present supplementary NMR data, which are necessary to verify that it's elucidated correctly. The presence of nitrogen wasn't demonstrated to me. The chemical elucidation of salsolinol needs to be discussed. Include high-resolution masses.
Author Response
The document was improved, but it doesn't present supplementary NMR data, which are necessary to verify that it's elucidated correctly. The presence of nitrogen wasn't demonstrated to me. The chemical elucidation of salsolinol needs to be discussed. Include high-resolution masses.
We apologize for not attending in the previous versions these important concerns. We came back to the original NMR data and retrieved the missing data. We apologize for the mistakes not corrected in the previous versions. The δ for C3 was missing in our table (40.1). This information is now included in Table 5. In addition, 2D COSY spectrum, HMBC-NMR and HMQC-NMR are now included in Supplemental figures 1-3. There was also a mistake in the Methodology: electron impact mass spectra were performed on a single equipment, HITACHI high-performance spectrophotometer. This was corrected in lines 278-280.
Reviewer 6 Report
Comments and Suggestions for Authors
All suggestions and comments are adequately included in a revised version of this article.
Round 3
Reviewer 4 Report
Comments and Suggestions for Authors
see PDF

Round 3
Authors have demonstrated that commercial Salsolinol displays biological activity. However, when they compared the 13C NMR data of the isolated compound from Senna silvestris with references 15 and 16, there was no concordance with C-3. Salsolinol must have 2 signals between 40 and 50 ppm, and this work indicates C-3 at 24.8 ppm.
Salsolinol This work Reference 15 Reference 16
C-1 50.4 52.0 52.4
C-3 24.8 40.5 40.9
This difference indicates that the naturally occurring compound must correspond with
another compound.
We apologize for not attending in the previous versions these important concerns. We came back to the original NMR data and retrieved the missing data. We apologize for the mistakes not corrected in the previous versions. The δ for C3 was missing in our table (40.1). This information is now included in Table 5. In addition, 2D COSY spectrum, HMBC-NMR and HMQC-NMR are now included in Supplemental figures 1-3. There was also a mistake in the Methodology: electron impact mass spectra were performed on a single equipment, HITACHI high-performance spectrophotometer. This was corrected in lines 278-280.
Reviewer 5 Report
Comments and Suggestions for Authors
Dear Authors, I believe you have fully complied with the request. I invite you to consider the above in your future research. Please correct the supplementary data in Figure 2; it shows COSY instead of HMBC.
Comments on the Quality of English LanguageNone
Author Response
Reviewer 5
Dear Authors, I believe you have fully complied with the request. I invite you to consider the above in your future research. Please correct the supplementary data in Figure 2; it shows COSY instead of HMBC.
We apologize for the mistake in Supplemental Figure 2, now corrected.
English accuracy
The manuscript has been edited for English accuracy.
Round 4
Reviewer 4 Report
Comments and Suggestions for Authors
See the PDF

Author Response
Reviewer 4
Round 4
Authors have corrected the 13C NMR data of the isolated compound from Senna silvestris
and they are in concordance with references 15 and 16. However, in figure 2 of the
supplementary material, they display a COSY experiment instead of HMBC experiment.
We apologize for the mistake in Supplemental Figure 2, now corrected.